# Advances in Somatic Embryogenesis of Banana

**DOI:** 10.3390/ijms241310999

**Published:** 2023-07-01

**Authors:** Mark Adero, Jaindra Nath Tripathi, Leena Tripathi

**Affiliations:** International Institute of Tropical Agriculture (IITA), Nairobi 30709-00100, Kenya; m.adero@cgiar.org (M.A.); j.tripathi@cgiar.org (J.N.T.)

**Keywords:** cryopreservation, embryogenic cell suspension, morphogenic genes, molecular mechanisms

## Abstract

The cultivation of bananas and plantains (*Musa* spp.) holds significant global economic importance, but faces numerous challenges, which may include diverse abiotic and biotic factors such as drought and various diseases caused by fungi, viruses, and bacteria. The genetic and asexual nature of cultivated banana cultivars makes them unattractive for improvement via traditional breeding. To overcome these constraints, modern biotechnological approaches like genetic modification and genome editing have become essential for banana improvement. However, these techniques rely on somatic embryogenesis, which has only been successfully achieved in a limited number of banana cultivars. Therefore, developing new strategies for improving somatic embryogenesis in banana is crucial. This review article focuses on advancements in banana somatic embryogenesis, highlighting the progress, the various stages of regeneration, cryopreservation techniques, and the molecular mechanisms underlying the process. Furthermore, this article discusses the factors that could influence somatic embryogenesis and explores the prospects for improving the process, especially in recalcitrant banana cultivars. By addressing these challenges and exploring potential solutions, researchers aim to unlock the full potential of somatic embryogenesis as a tool for banana improvement, ultimately benefiting the global banana industry.

## 1. Introduction

Bananas and plantains (*Musa* spp.) are important staple food crops grown in over 136 countries in the tropics, with global production of 170 million tons annually (FAOSTAT2021). It is a valuable food security and cash crop as it can be cultivated in diverse environments and produces fruits throughout the year in these favorable weather conditions. Smallholder farmers mainly cultivate banana for domestic consumption and local or regional markets; only about 15% of production enters international markets. Hundreds of cultivars of bananas are grown and consumed worldwide, but large-scale farmers mainly grow the Cavendish type of dessert bananas for commercialization in local and international markets. Other dessert banana cultivars, such as ‘Sukali Ndiizi’, ‘Gros Michel’, ‘Rasthali’, and ‘Lakatan’, are also produced in several countries worldwide, where they are sold for economic and social benefits. Plantain is grown largely in Central and West Africa, and East African Highland banana (EAHB) is cultivated in East Africa, especially in the Great Lakes region of Africa, and is a source of livelihood for millions of people [1,2].

Banana production is affected by several diseases and pests, declining soil fertility, narrow genetic diversity in germplasm, and inadequate availability of clean planting material among smallholder farmers. In addition, changes in climate and weather are also significantly impacting banana yields, particularly in the regions where the crop is grown with minimal or no irrigation. These factors reduce its production and threaten livelihoods. These cultivation and production constraints are exacerbated by the triploid nature of cultivated banana cultivars, which is a challenge to breeding programs and compromises the goal of expanding the genetic diversity of the crop [1,3]. Thus, banana improvement via modern biotechnology is important in ensuring their existence and improved yield. Biotechnological approaches to banana improvement include in vitro regeneration techniques involving organogenesis and somatic embryogenesis, which are prerequisites for further improvement via genetic engineering [4].

Organogenesis is a morphogenic response involving the de novo development of plant organs, such as shoot buds and roots, either directly from the explant or callus [5]. It is the basis for the in vitro propagation of bananas through axillary budding, which is a useful technique for providing disease-free and true-to-type planting materials for farmers and has been shown to improve yield significantly [6]. Somatic embryogenesis is a unique phenomenon in plants involving the development of embryos from somatic cells [7]. Like in most plant species, somatic embryogenesis in banana is induced by exogenous plant growth regulators (PGRs) in specific tissues, such as meristems and immature male flowers. Embryogenic calli from these tissues easily proliferate into embryogenic cell suspensions (ECS), which can be regenerated into millions of complete plantlets. ECS can be cryopreserved for long-term preservation. The ECS are also ideal explants for genetic engineering, including transgenesis and gene editing [8]. Thus, somatic embryogenesis is critical to banana improvement through modern breeding. However, somatic embryogenesis is highly dependent on genotype and very laborious. Only a few cultivars of bananas have so far responded to somatic embryogenesis, with most elite cultivars, including most EAHBs, remaining recalcitrant [9]. Notably, low frequencies of somatic embryogenesis are observed in the few cultivars amenable to the process. Therefore, there is a need to explore new approaches for inducing somatic embryogenesis in recalcitrant elite cultivars and improve the success rates of somatic embryogenesis in cultivars that show poor response to the process. Accumulating evidence shows that somatic embryogenesis in banana is modulated by certain genes, including transcription factors, induced by specific PGRs and environmental conditions [10,11]. This review discusses advances in somatic embryogenesis in banana with emphasis on the progress, molecular basis underlying the process, and the potential application of this knowledge to optimize somatic embryogenesis in recalcitrant banana cultivars.

## 2. Progress in Somatic Embryogenesis of Banana

The earliest records of somatic embryogenesis in banana can be traced to the 1980s when Cronauer and Krikorian obtained somatic embryos from cell suspensions of split young shoots [12]. Based on their findings, somatic embryos were only obtained in a medium containing 2,4,5-trichlorophenoxyacetic acid (2,4,5-T) instead of 2,4-dichlorophenoxyacetic acid (2,4-D) commonly used today. A year later, Escalant and Teisson [13], for the first time, reported successful plant regeneration from somatic embryos derived from the callus of zygotic embryos of diploid banana using 2,4-D. The same year, Novak and colleagues [14] regenerated hundreds of plants from somatic embryos derived from dicamba-induced callus of leaf sheaths and rhizome tissues. Progress on somatic embryogenesis and the establishment of embryogenic cell suspension cultures proceeded swiftly, and breakthroughs in 1991 were game-changing. Specifically, the first report on the use of multiple meristems (commonly known as scalps) as explant for inducing somatic embryogenesis and establishment of ECS [15]. In the same year, Ma described an elaborated method for establishing ECS from embryogenic calli derived from immature male flowers [16]. It is noteworthy that scalps and immature male flowers have been the most responsive and extensively used explants for establishing ECS. In 1993, Panis et al. [17] obtained and regenerated protoplasts from ECS derived from scalps. Other notable reports on the optimization of somatic embryogenesis and plant regeneration in the 1990s and early 2000 were the establishment of temporary immersion cultures and bioreactor systems to increase the efficiency of mass propagation of plants from ECS [18,19], and cryopreservation of banana ECS for long-term storage [20].

Notably, research in somatic embryogenesis of banana has not seen major discoveries in the last two decades. New reports have mainly focused on optimizing the previous protocols and processes in various banana and plantain cultivars. For example, Xu et al. [21] reported an efficient protocol for developing and regenerating ECS derived from scalps of banana cv. ‘Cavendish Williams’. Wong et al. [22] enhanced plant regeneration from ECS by incorporating a liquid-based embryo development medium. Further, Strosse et al. [23] conducted large-scale experiments to assess the embryogenic potential of scalps from 18 different banana cultivars belonging to five genome groups, including wild diploid (AA), Cavendish (AAA), highland (AAA-EAH), plantain (AAB), and cooking types (ABB). They confirmed that the scalp is a suitable explant for somatic embryo induction in banana, but success was not achieved with the highland banana cultivars. However, successful induction of somatic embryogenesis and ECS from immature male flowers of one cultivar of EAHB has been reported [24]. In 2007, Wei et al. [25] demonstrated that picloram could be used as an alternative to 2,4-D for the induction of somatic embryos in male flower explants of banana. The researchers also showed that a low concentration of thidiazuron (TDZ, 0.2 mg/L) could increase the frequency of somatic embryo germination. Torres et al. [26] revealed that meristematic domes of axillary sprouted buds from a diploid banana Calcutta 4 responded better than scalps for somatic embryogenesis. An efficient somatic embryo induction and regeneration protocol was optimized for the plantain cultivar ‘Gonja manjaya’ using scalps, where up to 30,000 plantlets were generated from 1 mL settled cell volume of ECS [27]. In 2014, Remakanthan et al. [28] reported unique findings in which direct somatic embryogenesis was induced from split shoot tips of banana cultivar ‘Grand naine’ in just 15 days, without an intervening callus phase. According to the study, combining 4.14 μM picloram and 0.22 μM benzyl aminopurine (BAP) led to 100% embryo induction; however, these findings have not been reproduced by any other group so far.

## 3. Genetic Stability of Banana Plants Derived from Somatic Embryos

Researchers have analyzed the genetic stability of plants regenerated from somatic embryos. Morais-Lino et al. [29] employed simple sequence repeat markers to analyze the genetic variations among banana plants regenerated from somatic embryos and found no genetic variation. Also, ISSR analysis of somatic embryo-derived banana plants did not detect soma-clonal variation among the plants, further confirming that somatic embryogenesis can serve as a tool for the large-scale regeneration of genetically stable plants [30].

## 4. Initiation and Steps of Plant Regeneration from ECS

Regeneration of banana plantlets via somatic embryogenesis involves several steps, including induction of embryogenic callus, initiation of ECS, embryo development, maturation, and embryo germination into complete plantlets, which are then acclimatized and potted (Figure 1a–j).

### 4.1. Induction of Embryogenic Callus

Somatic embryogenesis in banana can be induced from various explants, including mature/immature zygotic embryos, immature male flowers, female flowers, multiple meristems (scalps), rhizome slices, and corm tissues [8]. However, scalps (Figure 1a) and immature male flowers (Figure 1b) have been the most responsive, with consistent results across diverse banana genotypes. The plant growth regulators and media used for somatic embryo induction vary with the explant. Most studies have reported success for immature male flower explants when the MA1 medium is used; whereas the ZZ medium was found to be appropriate for scalps (Table 1) [16]. MA1 medium has four times higher 2,4-D concentration than ZZ medium, indicating that immature male flowers require more 2,4-D for callus induction than scalps. Variations in the type and concentrations of auxins have also been reported, particularly in the substitution of 2,4-D alone or with either picloram or dicamba for callus induction [14,25,31]. However, such variations have not been reproducible in most laboratories and banana genotypes. Regarding culture conditions, somatic embryogenesis in banana occurs better in the dark at 26–28 °C and mostly emerges after six weeks of culture, reaching the highest frequency at three months from the time of explant initiation in the callus induction medium. The ideal embryogenic callus (Figure 1c) for initiating ECS is typically obtained after four to five months of culture, particularly when immature male flowers are used as explants. The culture usually begins to turn brown after six months due to medium depletion and accumulation of phenolics. Thus, the timely selection of the ideal callus for transferring to a liquid medium is important for the successful initiation of ECS [32].

### 4.2. Initiation and Maintenance of Cell Suspension Cultures

Generation of ECS from embryogenic calli has contributed significantly to banana improvement. ECS are generated by culturing embryogenic calli in a liquid medium and maintaining the culture indefinitely in a rotary shaker. The components of the liquid media vary with the explant and medium used for callus induction. For example, calli induced from scalp explants proliferate easily in the same medium used for callus initiation minus a gelling agent (ZZ, Table 1) [23]. On the other hand, significant changes in the cell suspension medium are made for callus induced from immature male flowers. Specifically, the 2,4-D concentration is reduced from 4 mg/L to 1 mg/L and other auxins such as indole-3-acetic acid (IAA) and naphthaleneacetic acid (NAA) are excluded from the liquid medium (MA2, Table 1) [16]. Initiation of ECS involves two stages; in the first stage, a friable embryogenic callus is dropped into a culture vessel (such as a 50 or 100 mL conical flask) containing 3 to 10 mL medium, depending on the size of the callus complex. The liquid culture is then maintained at 26–28 °C in a shaker (95–100 rpm) in the dark for about two weeks or until the cells begin to proliferate rapidly. In the second stage, the suspension culture is transferred into a bigger vessel (such as a 250 mL conical flask), after which the culture medium is adjusted to about 50 mL (Figure 1d). The ECS in the bigger flask is maintained in the dark indefinitely at 26–28 °C. Half of the liquid medium is usually replaced with fresh medium every 7 to 20 days, depending on the rate of cell proliferation. The ECS cultures are usually well proliferated and ready for other uses at three months post-initiation and remain useful for at least nine months [32].

### 4.3. Development, Maturation, and Germination of Somatic Embryos

Embryogenic cells from cell suspension cultures need to develop into somatic embryos (Figure 1e) and then mature (Figure 1f) before they can germinate into shoots (Figure 1g) and then into plantlets (Figure 1h). This process typically takes four to six months, depending on the quality of ECS. The medium for embryo development, maturation, and germination also varies depending on the origin of the ECS [32]. Usually, ECS from scalp explants require a much simpler medium for development, maturation, and germination than those from immature male flowers. Specifically, ECS derived from scalp explants can develop, mature, and regenerate into plants in semi-solid half-strength Murashige and Skoog (MS) without PGRs (RD1, Table 1). However, a trace amount of BAP (0.25 mg/L) can be added to the RD1 medium to increase germination frequency after three months of embryo maturation [23,32]. Meanwhile, ECS derived from immature male flowers require richer media supplemented with several PGRs and antioxidants to mature and regenerate into plants (MA3 and MA4, Table 1) [32]. After germination, banana plants can be multiplied further in a proliferation medium (PM, Table 1, Figure 1i) supplemented with BAP [27]. Notably, banana shoots maintained in PM supplemented with BAP may not develop proper roots and must be transferred to a rooting medium (RM, Table 1) for complete plant regeneration [27].

### 4.4. Post Flask

Acclimatization is the final step before the plants can be introduced to the farm or greenhouses for further growth and development. Ex-vitro banana plants are easier to acclimatize than those from other plant species and do not require special substrates such as peat moss or vermiculite. Well-rooted plants (Figure 1i) can be transferred to normal soil in small pots after washing off the medium. However, the plants must be maintained in a humid environment for at least three weeks to ensure a high survival rate [35]. Upon acclimatization, the survival rate of banana plants is almost 100% for all the cultivars. Banana plantlets of cultivar ‘Grand naine’ transferred to soil after four weeks in rooting medium normally exhibit high survival rate (100%) during acclimatization [35]. Acclimatized plants are usually transferred to bigger pots (Figure 1j) for further use.

### 4.5. Cryopreservation of Cell Suspension Cultures

Cryopreservation is a crucial technique in plant biotechnology that ensures long term and safe storage of important germplasm [36]. It takes about 12–18 months to develop banana ECS. In addition, periodic refreshing of the culture medium is time-consuming, labor-intensive, and involves using chemicals and equipment, making it expensive to maintain ECS. Long-term culture and frequent handling of ECS also expose them to the risk of somaclonal variation, contamination, and reduced and eventual loss of morphogenic potential. Cryopreservation of banana ECS is, thus, a vital technique in laboratories that generate and maintain ECS. However, studies on the cryopreservation of banana ECS are limited. The most widely applied protocol for cryopreservation of banana ECS was described by Panis et al. [20]. The protocol was initially optimized for cryopreserving ideal ECS, which are homogenous, containing isodiametric cells with multiple small vacuoles, large nuclei, and tiny protein and starch granules. The procedure was later optimized to include less ideal ECS, including heterogeneous cell suspensions containing cells with large vacuoles and starch granules [37]. Generally, the protocol involves first subjecting ECS to cryoprotection using dimethyl sulphoxide at 0 °C, then slowly freezing them to −40 °C at the rate of 1 °C per minute before dipping them in liquid nitrogen for storage. After storage, the cells are thawed rapidly in a water bath at 40 °C. A protocol for cryopreservation of banana ECS via vitrification has also been reported [38]. As an alternative to cryopreservation, banana ECS can also be maintained for longer periods under slow growth conditions [39]. Notably, cryopreserved ECS has been shown to regenerate into normal plants with typical morphology and yield [40].

## 5. Factors Influencing Somatic Embryogenesis

Generating ECS of most elite banana cultivars remains a challenge because most of these cultivars are irresponsive to the available protocols. In the past, most somatic embryogenesis protocols for plants were developed based on a trial-and-error approach; however, increasing reports on somatic embryogenesis and recent advances in molecular biology uncover molecular mechanisms underpinning the process, providing novel research directions. Here, we summarize the factors to consider when optimizing somatic embryogenesis in banana.

### 5.1. Explant Selection and Manipulation

Most banana cultivars have not responded to somatic embryogenesis using the most responsive explants like scalps and immature male flowers. Thus, the focus should shift to explant selection and manipulation before callus initiation. Factors such as age and lushness affect the response of an explant to somatic embryogenesis. For instance, the response of immature male flowers to somatic embryo induction is better when the male buds are collected within three weeks after emergence, which also varies with the genotype [32,41]. Also, the position of immature flowers on the male bud has been shown to affect somatic embryogenesis [42]. The immature flowers from position 8–16 respond the best for callus induction. These factors could be attributed to the effect of various environmental and age-related factors on the expression of genes associated with somatic embryogenesis. Explant manipulation before in vitro culture may also affect somatic embryo induction and frequency. For instance, wounding of explants before culture can activate specific genes involved in somatic embryogenesis and improve explant response to the process [43]. Therefore, future studies should explore the effects of explant-related factors on somatic embryogenesis to improve the frequency of somatic embryogenesis in recalcitrant banana cultivars.

### 5.2. Plant Growth Regulators for Embryogenesis

Somatic embryogenesis is induced by supplementing the medium with different combinations of exogenous auxins and cytokinins. Typically, higher auxin concentration relative to cytokinin is needed for the induction and proliferation of somatic embryos in banana [8]. Despite the obvious participation of PGRs in somatic embryo induction, knowing the type, quantities, and combinations to apply for a specific banana genotype remains challenging, especially because endogenous levels of PGRs vary between genotypes. Thus, current studies still rely on the trial-and-error approach to optimize somatic embryogenesis. However, this is changing with recent molecular studies, which continue to uncover the intricacy of hormone signaling in plants. For example, two enzymes, indole-3-pyruvate monooxygenase and adenylate isopentenyl transferase, involved in endogenous auxin synthesis, were found to accumulate more in embryogenic than non-embryogenic callus of banana, suggesting that endogenous auxin levels influence somatic embryogenesis [44]. Thus, a further study designed to incorporate tryptophan in the medium, a precursor of IAA, reported enhanced somatic embryo induction and germination frequencies in banana cultivars that were initially recalcitrant to somatic embryogenesis. Also, the addition of NAA and gibberellic acid (GA), which are associated with the synthesis of enzymes that induce somatic embryo germination in banana, enhanced the germination of somatic embryos [45]. These findings suggest that formulating the right hormone regime could improve somatic embryogenesis in recalcitrant banana cultivars. However, further studies are needed to determine the exact hormone type and quantities for a specific genotype.

### 5.3. Culture Medium Additives

Plant tissue culture medium additives such as amino acids and antioxidants also play a significant role in somatic embryo induction and germination. Amino acids provide organic nitrogen and serve as precursors for endogenous synthesis of proteins and PGRs that facilitate somatic embryogenesis. Proline and glutamine are associated with embryo maturation, which improves embryo germination [42]. Tryptophan promotes embryogenesis by boosting endogenous auxin synthesis [45]. Antioxidants such as ascorbic acid, citric acid, and cysteine are crucial in reducing phenolic accumulation and explant necrosis [46,47]. Ethylene accumulation is associated with reduced somatic embryogenesis; thus, the addition of ethylene inhibitors such as salicylic acid can enhance somatic embryo induction [48]. Recently, the addition of calcium chloride in the callus induction medium significantly improved the frequency of somatic embryogenesis in recalcitrant banana cultivars [49]. Calcium-related proteins, including calcium-dependent protein kinase and the calcium-binding mitochondrial carrier protein, were upregulated in embryogenic callus more than non-embryogenic callus, suggesting that calcium potentially plays a crucial role in somatic embryogenesis [49]. However, further molecular analysis should be conducted to determine the precise role of these medium supplements in somatic embryogenesis.

### 5.4. Environmental Growth Conditions

Conditions under which plant tissue cultures are maintained significantly affect the outcome of an experiment and, therefore, should be considered when optimizing somatic embryogenesis in banana. These conditions mainly include temperature, light, and humidity but can be extended to include medium conditions such as gelling strength, pH, and frequency of sub-culture/medium change. Temperature regulates many physiological processes in plants and thus plays a crucial role in somatic embryogenesis. The optimum temperature for somatic embryogenesis can vary with genotype and the embryogenesis stage. For instance, enhanced somatic embryo induction was observed at a lower temperature of 17 °C than at 28 °C in radiata pine [50]. In contrast, a higher temperature of 30 °C is optimum for somatic embryo induction in cotton, while 25 °C is optimal for embryo maturation and germination [51].

Light requirement during somatic embryogenesis also varies with genotype and stage of embryogenesis [52,53]. In banana, somatic embryogenesis has been reported to occur better in darkness; however, further studies should be conducted to determine whether this is true for all genotypes. Studies have also revealed that somatic embryo induction varies with light quality. For instance, blue light improved the frequency of somatic embryo induction in *Agave tequilana* [54]. Moreover, a high ratio of red to far-red was found to promote somatic embryogenesis in *Araujia sericifera* [55], suggesting that light quality may play a significant role in somatic embryogenesis.

Other factors such as humidity, gelling strength, pH, and frequency of sub-culture also affect somatic embryogenesis. For example, Strosse et al. [23] reported that a relative humidity of more than 70% is optimum for somatic embryogenesis in banana.

## 6. Molecular Basis of Somatic Embryogenesis

Molecular studies have revealed specific genes and signaling pathways that regulate plant somatic embryogenesis. These genes comprise transcriptional factors, cell cycle proteins, and stress/defense-related proteins that modulate endogenous hormone synthesis and other signaling pathways [56]. Transcriptional factors play a critical role in the transcriptional regulation of somatic embryogenesis, and recent reports have implicated the roles of WUSCHEL (WUS), BABYBOOM (BBM), LEAFY COTYLEDON (LEC), and SOMATIC EMBRYOGENESIS RECEPTOR-LIKE KINASE1 (SERK1) in enhancing somatic embryo formation efficiencies [57].

WUS is a homeodomain TF modulating the formation of apical shoot meristem, which is crucial in somatic embryogenesis [58]. WUS expression is induced by auxins and mediates cell switch from a vegetative to an embryogenic state in *A. thaliana* [59]. Upregulation of WUS genes has been reported during somatic embryogenesis in several plant species, including *Medicago truncatula* [60], *Elaeis guineensis* [61], and *Betula platyphylla* [62]. Recently, overexpression of *GhWUS* from *Gossypium hirsutum* in *A. thaliana* promoted somatic embryogenesis [63]. A banana homolog of *WUS* (*MaWUS2*) is upregulated during the late stage of ECS, revealing that it potentially plays a crucial role in ECS proliferation in banana [10].

*BBM* gene encodes AP2/ERF domain transcription factor implicated in multiple plant morphogenic responses [64,65]. It also interacts with other TFs, such as LEC1/2, ABI3, and FUS3 network to induce somatic embryogenesis [66]. BBM has been implicated in somatic embryogenesis in many plant species, including *A. thaliana* [67], *Capsicum annuum* [68], *Glycine max* [64], and *Zea mays* [69]. Shivani et al. demonstrated that a banana homolog of *BBM* (*MaBBM2*) is highly expressed at all stages of ECS development relative to NECS, suggesting that it could play an essential role in somatic embryo induction and proliferation in banana [10].

The SERK family of TFs can serve as a marker of somatic embryogenesis in several plant species [70,71,72]. Initially identified in *Daucus carota*, SERK genes are upregulated during somatic embryo formation up to the globular stage, indicating their essential role in the initiation and early stages of somatic embryogenesis [72]. Analysis of *MaSERK1* expression in banana showed that it is upregulated in the embryogenic callus of immature male flowers compared with the non-embryogenic callus. Its expression is also higher during ECS proliferation, suggesting that it may be vital in the formation and proliferation of embryogenic callus in banana [73].

The LEC family of TFs plays significant roles in somatic embryogenesis and has been demonstrated to induce somatic embryo formation in heterologous systems [74,75,76,77]. However, the transcript level of its homolog (*MaLEC2*) in banana is higher in NECS, and it remains unclear whether *MaLEC2* plays a negative regulatory role in banana somatic embryogenesis [10].

Besides TFs, other genes that regulate somatic embryogenesis have been identified in banana. Kumaravel et al. identified many genes involved in somatic embryo induction, development, and germination. Specifically, calcium-binding mitochondrial carrier protein and calcium-dependent protein kinase (CDPK) were upregulated 34.2 folds in embryogenic callus relative to non-embryogenic callus of banana cv. Rasthali [49]. CDPK facilitates calcium transfer into the cell to promote cell differentiation and proliferation during somatic embryogenesis [78,79]. Shivani et al. studied the differential expression patterns of 16 genes during somatic embryogenesis and ECS proliferation in banana and identified *MaPIN1* as a positive regulator and *MaCRE2* and *MaCRE3* as either negative regulators or insignificant in banana somatic embryogenesis [11]. Furthermore, molecular analysis of genes expressed during somatic embryogenesis in banana identified tryptophan aminotransferase relate 2 (TAR2), an enzyme involved in auxin biosynthesis, as an important facilitator of the transition from early to late stages of somatic embryo development [80]. Defense-related proteins have also been implicated in somatic embryo induction in banana, including cysteine proteinase inhibitor, 5-epi-aristolochene synthase, and E3 ubiquitin-protein ligase [49]. Cysteine-rich peptides facilitate the supply of amino acids and nitrogen in germ tissues during germination and have been used as molecular markers for somatic embryo maturation [81,82]. Notably, higher levels of chaperon heat shock 70 protein were observed in EC than in NEC, suggesting that defense and stress-related proteins could also play an essential role in the somatic embryogenesis of banana [49]. In contrast, upregulation of enzymes related to ethylene biosynthesis (such as 1-aminocyclopropane-1-carboxylate oxidase) was observed in NEC of banana, implying that ethylene biosynthesis could suppress somatic embryo induction [49].

Apart from DNA sequence variations, epigenetic factors have recently emerged as significant contributors to somatic embryogenesis [83]. This effect is mainly associated with the degree of chromatin remodeling during somatic embryogenesis and could be influenced by explant choice and other environmental factors, including medium components and culture conditions [84]. Although the effect of epigenetic factors on somatic embryogenesis has not been demonstrated in *Musa* spp., insights from studies involving other plant species could inform future related studies in banana. Studies have shown that DNA methylation influences somatic embryogenesis [57]. Specifically, lower global DNA methylation has been reported in embryogenic cultures, while the reverse has been observed in non-embryogenic cultures of several plant species [85,86,87]. A study on *A. thaliana* demonstrated that de novo DNA methylation and its maintenance can affect somatic embryogenesis [88]. Notably, alterations in the pattern of global DNA methylation, especially during long-term subcultures in exogenous PGRs, can affect the embryogenic potential of pro-embryogenic masses [89]. Many experiments have been conducted to demonstrate that DNA methylation indeed affects somatic embryogenesis. For example, the application of the demethylating agent 5-azacitidine (5AzaC) inhibited somatic embryogenesis in *A. thaliana* and other plant species [88,90,91]. Posttranslational modification of histone has also been implicated in somatic embryo formation [92]. Histone deacetylation modulates cell cycle reprogramming during the initial stages of somatic embryogenesis, which could be associated with the presence of exogenous auxins [93]. Also, the expression of polycomb repressive complex 2 (PRC2), a gene family involved in lysine 27 methylation in histone H3 is negatively associated with somatic embryogenesis and is explant specific [94].

## 7. CRISPR/Cas9 Based Genome Editing of Banana Using Embryogenic Cell Suspension

Genome editing via CRISPR/Cas9 has recently gained significance as the method of choice for genome engineering of crops. CRISPR/Cas9 tools have been recently optimized for banana and plantain crops using embryogenic cell suspension in several laboratories. A CRISPR/Cas9-based genome editing system for bananas has been established, targeting the knockout of the phytoene desaturase (PDS) as a visual marker gene [95]. The endogenous banana streak virus (eBSV) integrated into the B genome of plantain (AAB) was inactivated using CRISPR/Cas9-based editing to overcome a key barrier in breeding and the distribution of hybrids [96]. BSV is a dsDNA badnavirus, whose genome gets integrated into the B genome of plantain. The gene-edited plantain “Gonja Manjaya” had targeted mutations in the eBSV sequences integrated into the host genome. Phenotyping of the edited events verified the inactivation of eBSV for its ability to generate functional infectious viral particles. Recently, it was demonstrated that the editing of *MusaDMR6* in banana using CRISPR/Cas9-mediated gene editing resulted in enhanced resistance to BXW disease [97].

In addition, the CRISPR/Cas9 was used to generate β-carotene-enriched Cavendish banana cultivar “Grand Naine” [98]. Further, CRISPR/Cas9 technology was applied to generate semi-dwarf banana cultivar “Gros Michel” by manipulating the gibberellin 20ox2 (MaGA20ox2) gene, disrupting the gibberellin (GA) pathway [99]. Recently, Hu et al. [100] demonstrated that editing aminocyclopropnae-1-carboxylase oxidase (MaACO1) in banana extended shelf-life through reduced ethylene synthesis.

## 8. Future Prospects for Improving Somatic Embryogenesis

The main obstacle hindering banana improvement via modern biotechnology is the recalcitrance of elite cultivars to somatic embryogenesis. Although manipulation of explants, medium components, and culture conditions can improve the frequency of somatic embryogenesis, it is improbable that such approaches will work for the most recalcitrant cultivars like the EAHBs. In addition, optimizing somatic embryogenesis via medium variation is time-consuming and less precise since it relies on trial-and-error approaches. Studies have also revealed that although many genes that mediate somatic embryogenesis are shared across diverse plant species, some are limited to certain plant species [57]. Thus, future breakthroughs in somatic embryogenesis, especially for the recalcitrant banana cultivars, will likely involve the heterologous expression of morphogenic genes. So far, studies on the heterologous expression of morphogenic genes have mainly focused on TFs, including *LEC*, *WUS*, and *BBM.* Although not yet reported in *Musa* spp., the overexpression of these TFs can significantly promote somatic embryogenesis and transformation efficiencies in diverse plant species, including dicotyledonous and monocotyledonous plants.

For example, *LEC 2* overexpression induced somatic embryogenesis in the seedlings of *A. thaliana*, confirming its potential role in auxin signaling [74]. In coffee, overexpressing *WUS* induced callus formation and somatic embryogenesis by 400 folds [101]. The heterologous expression of *AtWUS* from *A. thaliana* in cotton enhanced callus differentiation and positively modulated other TFs, including *FUS3*, *LEC1*, and *LEC2* [102]. In addition, ectopic expression of BBM in *A. thaliana* revealed that it is mainly upregulated in plant tissues undergoing differentiation and morphogenesis [103]. Heterologous expression of *BBM* from *Brassica napus* and *A. thaliana* in *Nicotiana tabacum* promoted cell differentiation. Still, they exhibited different responses depending on the plant species, suggesting that components of its signaling pathway may vary between species [104]. Notably, somatic embryogenesis without hormone supplementation in the culture medium was observed in transgenic lines of *A. thaliana* and *Theobroma cacao* overexpressing *the TcBBM* gene from *T. cacao*. However, subsequent suppression of embryo development was observed in the *T. cacao* transgenic lines, implying that overexpression of *BBM* genes can adversely affect some genotypes’ later stages of somatic embryogenesis [105]. Recent monocot studies have shown that ectopic expression of morphogenic genes can expand the genotype range for somatic embryogenesis and transformation. For instance, overexpression of maize *BBM* and *WUS* genes enhanced somatic embryogenesis and transformation of recalcitrant maize, sorghum, and rice cultivars [106,107,108]. Genome-wide analysis of TFs during somatic embryogenesis in banana revealed that *MaWUS2* and *MaBBM2* could play a crucial role in somatic embryogenesis, suggesting that their overexpression has a great potential in revolutionizing somatic embryogenesis and transformation of recalcitrant banana cultivars [10]. 

## 9. Conclusions

Somatic embryogenesis and the establishment of ECS in banana are highly dependent on genotype, which has limited banana genetic engineering for decades. Moreover, the process is time-consuming, and the maintenance of ECS is expensive and laborious. As a result, only a few banana cultivars have seen significant advancements through this method, while elite cultivars like the EAHBs have been left behind. Somatic embryogenesis in banana can be optimized by manipulating explants, culture medium components, and growth room conditions. However, overcoming the challenges posed by highly recalcitrant cultivars, particularly EAHB requires additional approaches. Fortunately, ongoing molecular studies provide new insights into the process and will likely contribute to future breakthroughs. Notably, recent investigations on the heterologous expression of morphogenic genes in monocots offer promising insights. This approach is anticipated to play a critical role in overcoming the genotype barrier associated with somatic embryogenesis and ultimately enabling the genome engineering of banana.

By leveraging these molecular advancements, researchers can work towards improving the efficiency and effectiveness of somatic embryogenesis in bananas. It could lead to the successful somatic embryogenesis of elite banana cultivars like the EAHBs, which were previously challenging to incorporate into biotechnology programs. The enhanced somatic embryogenesis has the potential to revolutionize banana improvement efforts through modern biotechnology and open new possibilities for improving the traits and characteristics of this globally important crop.

## Figures and Tables

**Figure 1 ijms-24-10999-f001:**
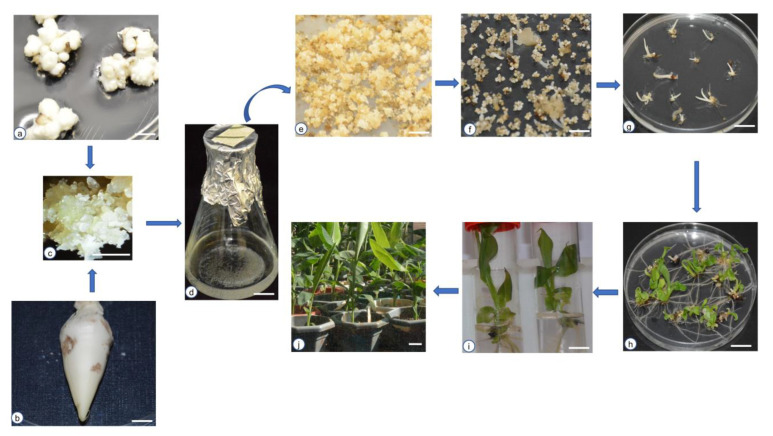
Steps of somatic embryogenesis and plant regeneration in banana. (**a**) Multiple meristems (also known as scalps) explants cultured for callus induction, (**b**) Male bud cultured for callus induction, (**c**) Friable embryogenic ideal callus, (**d**) Embryogenic cell suspension, (**e**) Embryogenic calli on embryo development medium, (**f**) Embryogenic calli on embryo maturation medium, (**g**) Embryos germinating in the dark, (**h**) Embryos germinating into plantlets and shoots turning green upon transferring them to light, (**i**) Fully developed plantlets with roots, (**j**) Complete plants transferred to pots in the greenhouse. Scale bar: 1 cm in (**a**,**b**,**d**–**i**), 10 cm in (**j**), and 5000 µm in (**c**).

**Table 1 ijms-24-10999-t001:** Medium used for banana somatic embryogenesis and regeneration.

Medium	Components	Step of Regeneration	Reference
MA1	Full-strength MS salts, MS vitamins, 3% sucrose, 1 mg/L biotin, 4 mg/L 2,4-Dichlorophenoxyacetic acid (2,4-D),1 mg/L naphthalene acetic acid (NAA), 1 mg/L indole-3-acetic acid (IAA), 3 g/L gelrite, pH 5.7	Callus induction	[16,27]
MA2	Full-strength MS salts plus vitamins, 3% sucrose, 1 mg/L biotin, 1 mg/L 2,4-D, 100 mg/L glutamine, 100 mg/L malt extract, pH 5.3	Cell suspension	[16,23]
MA3	SH basal salts, MS vitamins, 60 mg/L ascorbic acid, 100 mg/L glutamine, 250 mg/L proline, 100 mg/L malt extract, 400 mg/L cysteine, 60 mg/L citric acid, 1 mg/L biotin, 0.2 mg/L NAA, 0.2 mg/L 6-(γ,γ-Dimethylallylamino) purine 2-ip, 0.2 mg/L kinetin, 0.1 mg/L zeatin, 4.5% sucrose, 3 g/L gelrite, pH 5.8	Embryo development	[16,23]
MA4	Full-strength MS salts, 3% sucrose, 2 mg/L IAA, 0.5 mg/L benzylaminopurine (BAP), Morel vitamins, pH 5.8	Embryo maturation and germination	[16,23]
RD1	½ strength MS macro salts, MS micro salts, 3% sucrose, 10 mg/L Ascorbic acid	Embryo maturation	[16,23]
RD2	½ strength MS macros salts, MS micro salts, 3% sucrose, 10 mg/L Ascorbic acid, 0.25 mg/L BAP, pH 5.8	Embryo germination	[23]
ZZ	½ strength MS macro salts, MS micro salts, 3% sucrose, 10 mg/L Ascorbic acid, 1mg/L 2,4-D, 0.2 mg/L zeatin, 3 g/L gelrite, pH 5.8	Callus induction and cell suspension	[23,27]
PM	Full-strength MS salts plus vitamins, 10 mg/L ascorbic acid, 3% sucrose, 2.5 mg/L BAP, 2.4 g/L gelrite, pH 5.8	Shoot elongation	[27]
RM	Full-strength MS salts plus vitamins, 10 mg/L ascorbic acid, 3% sucrose, 1 mg/L indole-3-butyric acid (IBA), 2.4 g/L gelrite, pH 5.8	Rooting	[27]

MS; Murashige and Skoog [33], SH; Schenk and Hildebrandt Medium [34].

## Data Availability

Not applicable.

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
