# Peer review of "Advances in Somatic Embryogenesis of Banana"

_ijms, 2023, doi:10.3390/ijms241310999_

Round 1
Reviewer 1 Report
The reviewed article provides a comprehensive overview of the challenges faced in banana cultivation and the importance of somatic embryogenesis as a biotechnological approach for improving banana varieties. The text touches upon the potential factors that could influence somatic embryogenesis, indicating that the review goes beyond a simple description of advancements and also delves into the challenges associated with the process. This adds depth to the review and highlights the importance of understanding the limitations and prospects of somatic embryogenesis in banana improvement.
The text is well-organized and understandable. I only have a few minor comments, that require clarification.
- Keywords should be arranged alphabetically. Moreover, please, do not repeat words from the title.
- All abbreviations must be explained when first mentioned.
- Please, follow the MDPI’s formatting style.
- Names of varieties should be written in ‘ ’.
- The authors incorrectly use the term ‘variety’ (wild). It should be ‘cultivar’ (created in a breeding process).
- “Organogenesis is a morphogenic response involving the development of plant organs” should be “Organogenesis is a morphogenic response involving the de novo development of plant organs”.
- Change ‘hormones’ to ‘plant growth regulators’.
- Please, provide the initials of the author of the species’ name when mentioning it for the first time.
- Figure 1 lacks scale bars.
- In the part about cryopreservation, some recent references are missing (e.g. https://doi.org/10.3390/biology11060847).
After making all the necessary changes, the article can be accepted for publication.
The quality of English is OK.
Author Response
We are grateful for your constructive feedback on the manuscript. We have adopted your suggestions and corrected the manuscript with track changes.
- Keywords should be arranged alphabetically. Moreover, please, do not repeat words from the title.
We have replaced some keywords and arranged them in alphabetical order as suggested. The new keywords are as follows: ‘Cryopreservation; Embryogenic cell suspension; Morphogenic genes; Molecular mechanisms’
- All abbreviations must be explained when first mentioned.
Thanks for your keen observation. This has been done.
- Please, follow the MDPI’s formatting style.
Thanks for your suggestion. The manuscript has been formatted according to MDPI’s formatting style.
- Names of varieties should be written in ‘ ’.
Thanks for your suggestion. This has been done.
- The authors incorrectly use the term ‘variety’ (wild). It should be ‘cultivar’ (created in a breeding process).
Thanks for your suggestion. This has been revised.
- “Organogenesis is a morphogenic response involving the development of plant organs” should be “Organogenesis is a morphogenic response involving the de novo development of plant organs”.
Thanks for your suggestion. The phrase ‘de novo’ has been added to the specific section.
- Change ‘hormones’ to ‘plant growth regulators’.
Thanks for your suggestion. This has been done.
- Please, provide the initials of the author of the species’ name when mentioning it for the first time.
Thanks for your suggestion. This has been done.
- Figure 1 lacks scale bars.
Thanks for your keen observation. Scale bars have been added to the figure.
- In the part about cryopreservation, some recent references are missing (e.g. https://doi.org/10.3390/biology11060847
Thanks for your suggestion. A recent reference has been added to the subsection.
Reviewer 2 Report
Dear authors
as a reviewer, I feel that this manuscript can be resubmitted after major improvements. Please use the template MDPI for this manuscript. The scientific quality of this manuscript must be improved.
Author Response
We are grateful for your constructive feedback on the manuscript. We have adopted your suggestions and corrected the manuscript with track changes.
We have formatted the manuscript using the MDPI template. The manuscript has also been improved.
Reviewer 3 Report
Comments and Suggestions for Authors
The presented article is a review of the progress in somatic embryogenesis of bananas. It focuses on the different steps involved in this process, cryopreservation techniques and the molecular mechanisms underlying the process, highlighting its progress. Factors that could influence somatic embryogenesis and prospects for improving the process are discussed, with emphasis on recalcitrant banana cultivars and the potential application of this knowledge to optimize somatic embryogenesis in recalcitrant banana cultivars.
The manuscript is composed according to the requirements of “International Journal of Molecular Sciences” for a review preparation. The review made on molecular advancements in somatic embryogenesis techniques of bananas will be useful for researchers to work towards improving the efficiency and effectiveness of somatic embryogenesis in bananas. It could lead to the successful somatic embryogenesis of elite banana cultivars like the EAHBs, which were previously challenging to incorporate into biotechnology programs.
The following recommendations can be made:
Introduction:
I suggest making a small correction in the following paragraph /last sentence on page 2 and the first 2-3 sentences on page 3/:
“Thus, banana improvement via modern biotechnology is important in ensuring their existence and improved yield. Biotechnical approaches to banana improvement include in-vitro regeneration techniques involving organogenesis and somatic embryogenesis, which are prerequisites for further improvement via genetic engineering [4].
Organogenesis is a morphogenic response involving the development of plant organs, such as shoot buds and roots, either directly from the explant or callus [5]. Thus, organogenesis is the basis for the in-vitro propagation of bananas. Micropropagation of banana is useful in providing diseasefree and true-to-type planting materials for farmers and has been shown to improve yield significantly [6].
Somatic embryogenesis is a unique phenomenon in plants involving the development of embryos from somatic cells [7].”
The corrections include some translocation and omission as follow:
“Thus, banana improvement via modern biotechnology is important in ensuring their existence and improved yield. Biotechnical approaches to banana improvement include in-vitro regeneration techniques involving organogenesis and somatic embryogenesis
Micropropagation of banana is useful in providing disease free and true-to-type planting materials for farmers and has been shown to improve yield significantly [6].These techniques are also a prerequisite for further improvement via genetic engineering [4].
Organogenesis is a morphogenic response involving the development of plant organs, such as shoot buds and roots, either directly from the explant or callus [5].
Somatic embryogenesis is a unique phenomenon in plants involving the development of embryos from somatic cells [7].”
An incorrect citation of a literary source has been noted in the text - the source number should be placed after the author's name and not at the end of the sentence:
Page 4:
“ … Cronauer and Krikorian obtained somatic embryos from cell suspensions of split young shoots [12]” The correct: Cronauer and Krikorian [12] obtained …
“…Escalant and Teision, for the first time, reported successful plant regeneration from somatic embryos derived from the callus of zygotic embryos of diploid banana using 2,4-D [13]. The correct: Escalant and Teision [13], for the first time …
“The same year, Novak and colleagues regenerated hundreds of plants from somatic embryos derived from dicamba-induced callus of leaf sheaths and rhizome tissues[14]. The correct: … Novak and colleagues [14] …
“…Novak and colleagues regenerated hundreds of plants from somatic embryos derived from dicamba-induced callus of leaf sheaths and rhizome tissues [14].” The correct: Novak and colleagues [14] …
“…Panis et al. obtained and regenerated protoplasts from ECS derived from scalps [17].” The correct: Panis et al.[17] obtained …
Page 5:
“Torres et al. revealed that meristematic domes of axillary sprouted buds from a diploid banana Calcutta 4 responded better than scalps for somatic embryogenesis [26]” The correct: Torres et al. [26] revealed that …
“Further, Strosse et al. conducted large-scale experiments to assess the
embryogenic potential of scalps from 18 different banana cultivars belonging to five genome
groups, including wild diploid (AA), Cavendish (AAA), highland (AAA-EAH), plantain (AAB), and cooking types (ABB) [23].” The correct: Strosse et al. [23] conducted …
“For example, Xu et al. reported an efficient protocol for developing and regenerating ECS derived from scalps of banana cv. Cavendish Williams [21].” The correct: Xu et al. [21] reported ….
“Wong et al. enhanced plant regeneration from ECS by incorporating a liquid-based embryo
development medium [22].” The correct: Wong et al.[22] enhanced …
Page 6:
“Morais-Lino et al. employed simple sequence repeat markers to analyze the genetic variations among banana plants regenerated from somatic embryos and found no genetic variation [29]” The correct: Morais-Lino et al.[29] …
Page 16:
“…Strosse et al. 2003 reported that a relative humidity of more than 70% is optimum for somatic embryogenesis in banana[23].” The correct: Strosse et al. [23] reported …
In conclusion, this manuscript is recommended for publication in the Special Issue: Recent Advances in Plant Somatic Embryogenesis: Where We Stand and Where to Go? Of “International Journal of Molecular Sciences”.
Author Response
We are grateful for your constructive feedback on the manuscript. We have adopted your suggestions and corrected the manuscript with track changes.
- I suggest making a small correction in the following paragraph /last sentence on page 2 and the first 2-3 sentences on page 3/:
Thanks for your suggestion. We have revised the sentence to improve its clarity as follows:
Organogenesis is a morphogenic response involving the de novo development of plant organs, such as shoot buds and roots, either directly from the explant or callus. It is the basis for the in-vitro propagation of bananas, a useful technique for providing disease-free and true-to-type planting materials for farmers and has been shown to improve yield significantly.
- An incorrect citation of a literary source has been noted in the text - the source number should be placed after the author's name and not at the end of the sentence:
Thanks for your keen observation. This has been corrected in the manuscript as suggested.